

# The effect of advocacy on perceived credibility of climate scientists in a Dutch text on greening of gardens

Erik van Sebille[1], Celine Weel[1], Rens Vliegenthart[2], Mark Bos[1]

[1]Freudenthal Institute, Utrecht University, Utrecht, 3584CC, Netherlands

[2] Strategic Communication Group, Wageningen University, Wageningen, Netherlands

*Correspondence to*: Erik van Sebille (E.vanSebille@uu.nl)

**Abstract**

Many climate scientists refrain from advocacy and activism because they worry it decreases their credibility. Through a survey of almost 1,000 Dutch respondents, we compare responses to a text written in a neutral tone to those of a text written in an advocating tone on perceived credibility of the authoring scientist in these texts. Analyses show that the perceived credibility of the scientist who authored the text increases by advocacy overall, and that the advocating scientist is considered more credible than the neutral scientist specifically in their perceived sensitivity and care for society. We also analyse the effect of

the type of visual element in the text, to test whether a visual element that is more science-based can increase the perceived credibility of the scientist in the knowledge domain. However, we do not find any significant differences between a scientific bar chart and a stock photo. Based on these results, we conclude that advocacy can increase the climate scientist's average perceived credibility. However, we find that the fraction of respondents that feels called to action is not higher for those who read the advocacy text, suggesting that advocacy does not stimulate behavioural change in this case.

**1 Introduction**

Given the urgency of the climate crisis, climate scientists are increasingly called upon to engage in advocacy as they may play a role in encouraging and inspiring the public to contribute to climate action, or to influence policymaking (Besley and Dudo, 2017; Capstick et al., 2022; Gundersen et al., 2022; Wijnen et al., 2024). A recent global study (Cologna et al., 2025) showed that indeed, in almost all countries included in the sample, a majority of people want scientists to be involved in policymaking.

In addition, a recent survey (Dablander et al., 2024) indicated that 29% of scientists engage in advocacy, and 58% of scientists are willing to do so. However, fear of negative consequences of engaging in advocacy stops many scientists from speaking out. Some scientists fear that advocacy undermines their credibility in the eyes of the public (Cologna et al., 2021; Dablander et al., 2024; Messling et al., 2025) because it is not the scientist's place to engage in advocacy, and they need to "let the science speak for itself" (Fischhoff, 2007). Moreover, scientists might fear that advocacy by one individual scientist may influence

how the public appreciates the scientific community as a whole (Kotcher et al., 2017). Yet, multiple studies among citizens in various countries revealed that the credibility of climate scientist as perceived by the public are not negatively affected by openly supporting climate policies (Cologna et al., 2021) or only in cases in which a scientist advocates for a very specific or highly controversial policy (Beall et al., 2017; Kotcher et al., 2017).



Previous research on reception of advocacy among scientists by the public has been either very general – in the sense of: "How do you feel about climate scientists advocating for specific policies?" (Cologna et al., 2021) – or focused on hypothetical social media content presented to the participants (Kotcher et al., 2017). There has thus been little research on platforms on which scientists and the public interact, such as question-and-answer (Q&A) platforms. Interaction on these platforms may be initiated by either party, instead of only providing 'downstream' communication from the scientific community to the public. For example, a member of the public might post a question related to climate change that a scientist may respond to, opening op dialogue between scientists and the public, which experts argue is the way forward to promote trust in scientists (Cologna et al., 2025; Gundersen et al., 2022). Furthermore, the questions and answers on these websites often pertain to very practical solutions that readers can take themselves, in their own daily lives.

The aim of this study is to explore how readers experience advocacy in a text on climate change impacts and solutions of the greening of gardens taken from such a Q&A-website, and to investigate whether advocacy affects how credible the readers consider the authoring scientist to be. We focus on a text about the urban climate effects of greenery in gardens as opposed to stone/tiled gardens (e.g., Heusinkveld et al., 2014; Klok et al., 2019), because the topic is relatively uncontroversial and because it facilitates discussion of both individual behaviour change (you should include more greenery in your garden) as well as policy-change (the government should oblige people to include more greenery in their gardens).

Learning more about the extent to which advocacy influences scientists' credibility as perceived by their readers may help scientists communicating climate change to determine which role they should take in terms of advocacy to promote sustainable behaviour while still being considered credible. Empirical findings from this study may serve as endorsement for the many climate scientists who are willing to take a more advocacy-driven approach in their communications but are unsure of the consequences (Dablander et al., 2024).

## 2 Theoretical background

### 2.1 Credibility in texts

The extent to which climate scientists should get involved in public engagement is discussed elaborately – and given the urgency of the climate crisis, scientists are increasingly called upon to speak out (Nelson and Vucetich, 2009; Besley and Dudo, 2017; Capstick et al., 2022; Gundersen et al., 2022; Büntgen, 2024; van Eck et al., 2024). In his framework, Pielke Jr (2007) defined four roles scientists may take in the context of policy: Pure Scientist, Science Arbiter, Issue Advocate, and Honest Broker of Policy Alternatives. In the context of this study, the roles of the Science Arbiter and the Issue Advocate will be the focus: one in which a scientist is asked a question and answers with only facts (Science Arbiter) and one in which the scientist advocates for specific choices (Issue Advocate).

An argument that is often used against advocacy by scientists is that it may hurt public trust in science (Büntgen, 2024; Fischhoff, 2007) or the perceived credibility of science in general or individual scientists (Cologna et al., 2021; Dablander et al., 2024). This would be problematic, as credibility has been found to be a strong mediator between information intake and





intended climate action (Dong et al., 2018), because a higher level of perceived credibility of the communicating scientists increases the readers' risk perception. Similarly, Attari et al. (2016) found that differences in perceived credibility of the author had a large effect on the participants' reported intentions to alter their own energy consumption. To change the reader's behaviour, the author must be perceived to be credible.

Concerning the public trust in science, a recent global study showed that scientists' competence, goodwill and integrity are perceived to be high (characteristics that are all connected to credibility), and that in nearly all countries most people want scientists to be involved in policymaking (Cologna et al., 2025). Additionally, public trust in science in the Netherlands is high (Van den Broek-Honingh et al., 2021) though there are also signs of increasing polarization in trust (Edelman Trust Institute, 2024). This shows that scientists may have quite some 'leeway' in terms of credibility, but still, over half of scientists globally

do not engage in advocacy, despite their willingness (Dablander et al., 2024).

   Previous studies have revealed that, indeed, public trust in scientists can be affected negatively when scientists adopt a persuasive tone of voice as opposed to an informative one, but that this decline in trust can be explained by a mismatch between the expectations the public has of the communicating scientist or organization and the actual communication (Rabinovich et al., 2012). In other words, when the scientist is transparent about their motivations, this decline in trust is not observed.

Adding to this, Cologna et al. (2021) found that 74% of members of the general public in the US and 70% in Germany believe climate scientists should actively advocate for specific climate-related policies. The researchers found that such active advocacy decreases a hypothetical scientist's perceived objectivity, but not their trustworthiness or honesty, and increases the public's perception of them acting in the interest of society. Cologna et al. (2021) concluded that their "results suggest that scientists' anxieties about loss of credibility from engagement may be misplaced" (p.8) and that scientists should not abstain

from public engagement based on fears of jeopardized credibility.

   Investigating the applicability of this conclusion in a Dutch context, we will focus on answering the following question (see also our pre-registration at https://aspredicted.org/w3cf-qgf9.pdf):

*RQ1: How does a text written in the Issue Advocate role versus the Science Arbiter role affect the credibility of the writing scientist?*

Based on outcomes of previous research (Beall et al., 2017; Kotcher et al., 2017; Cologna et al., 2021), the hypothesis to this question is as follows:

*H1: Readers of a text on climate change impacts and solutions of greening gardens will perceive an authoring scientist taking either the Issue Advocate or the Science Arbiter role as equally credible.*

## 2.2 Credibility in visual elements

Furthermore, we will investigate whether the type of visual element that is included with the article has an impact on the credibility. In past research there has been a lot of attention for written science communication, but less for the role of visuals (Murchie and Diomede, 2020). There are indications that visuals used to communicate science in general and climate change specifically are important (León et al., 2022; McCabe and Castel, 2008). Visuals play a critical role in environmental



communication, influencing both cognitive and affective responses (Borah, 2009; Rodriguez and and Dimitrova, 2011).
Visuals are perceived in an associative and quick manner and are better at attracting attention than text (Mooseder et al., 2023). Moreover, they have the ability to quickly convey the general gist of a message (Ware, 2008). Therefore, visuals are considered highly relevant and influential message features that can be expected to play a role in the reach and impact of messages (Li and Xie, 2020). In the context of climate change specifically, research has shown that specific visuals, such as visuals that tell stories, include local connections, and show "real" people,  affect the levels of concern and that they may play a role in
promoting public engagement with the issue (León et al., 2022; Metag, 2020). Indeed, Smith and Leiserowitz (2014) argued that environmental campaigns can leverage the power of visuals to raise awareness, evoke affective responses, and motivate people to take action. Visuals have the potential to emphasize the severity of the issue and make climate change feel more concrete (Wang et al., 2018). More recently, Li et al. (2023) showed more artistic visualizations, when compared to data graphs, elicited stronger positive emotions but did not differ in perceived credibility or effectiveness.

However, the precise effects of visuals in a multimodal setting with texts remain obscure. Leerink et al (2024) suggest that the limited effect of personalization on the credibility of an author that they found in their experiment could be because the visual element (a graph) was unchanged, "giving the overall look of the article a more expository feel." In an academic context, Pferschy-Wenzig et al. (2016), for example, showed that the mere presence of a graphical abstract in a paper does not automatically lead to more engagement with that paper (higher rates of article downloads, abstract views, or citations). Ibrahim
et al. (2017), however, showed that graphical abstracts in social media posts were related to higher engagement with these posts. And in the context of pro-environment communication, Lazard and Atkinson (2015) also showed that visuals such as infographics are effective tools to communicate messages intended to change attitude and behavior and recommend the evaluation of the use of visuals in other contexts.

There are many examples of visual communication in science, but these mostly focuses on graphs and figures in academic
interactions (Rodríguez Estrada and Davis, 2015). And while data visualizations such as graphs can be effective, they can also be misleading (Szafir, 2018; Wijnker et al., 2022). This duality is the result of our natural and intuitive mode of information processing – e.g., bigger means more, closer means related, etc. (Broek, 2012; Cairo, 2019). As León et al. (2022) argued, in these screen-based and graphics-heavy times, understanding the role visuals play in science communication is especially important.

To explore to what extend a scientific visual will increase credibility of the scientist, especially in the competence domain, we will also research the following secondary research question:

*RQ2: How does a visual element that shows scientific data affect the credibility of the authoring scientist, compared to a more general stock photo?*

As we expect that visual elements that show scientific data increase the credibility of the author, our associated hypothesis is:
*H2: Readers of a text on climate change impacts and solutions of greening gardens will perceive authors as more credible when the visual element shows scientific data.*



## 3 Methods

### 3.1 Context

This research uses an original text from the Q&A platform KlimaatHelpdesk, which is a Dutch website on which visitors
submit questions that are answered and peer-checked by scientists or experts (Szafir, 2018). Questions and answers are
published on the platform, and all visitors can access the previously asked and answered questions. The pool of experts who
provide answers includes over 400 Dutch speaking contributors, representing a wide array of disciplines; similarly, questions
on KlimaatHelpdesk include a wide range, from individual behaviour ("Which is better for the climate: a paper book or an e-
book?") to geographical explanations ("What is the impact of extreme weather, caused by climate change, on nature?") to
societal impact ("How do we get politics in motion to combat the climate crisis?") to name a few. KlimaatHelpdesk's objectives
are to provide "high-level scientific findings in a simple language that is accessible for the general public" and "answers based
on scholarly standards of scientific integrity and objectivity" (Stichting KlimaatHelpdesk, 2023a). In 2023, KlimaatHelpdesk
attracted 98,000 visitors and answered 32 questions (Stichting KlimaatHelpdesk, 2023b). Note that KlimaatHelpdesk provides
limited possibility for interaction; a visitor poses a question, which is answered by an expert. There is no room for interaction
after this on the Q&A platform itself. However, the Q&A characteristic of KlimaatHelpdesk makes it possible to distinguish
Pielke's roles of Science Arbiter and Issue Advocate.

### 3.2 Text conversion

We select an existing Dutch text from KlimaatHelpdesk with the (translated) title: "What are the climate effects of a tiled or
green garden?" The original Dutch text is shortened for the Science Arbiter condition, from 1369 words to 331 words, and the
title is changed to "Removing all tiles from Dutch gardens is good for climate". To make the original author unrecognisable,
the byline of the original text is changed to the non-existing "Prof Ben van Weel", with an affiliation as professor of ecology
and climate at Utrecht University.

Then, the text is adjusted by adding advocacy elements to form the Issue Advocate condition text. Concrete guidelines on
writing as an *Issue Advocate* are sparse. Multiple experts agree that stating what one 'should' do or prefer is a key characteristic
for advocacy (Pielke Jr, 2007; Donner, 2014) and that scientists should only address issues that are within their scientific
expertise (Gundersen et al., 2022; Pierson, 2012; Steneck, 2011). Additionally, Pierson (2012) and Steneck (2011) agree that
advocating scientists should point out limitations, address opposing scientific views when relevant, and explaining the
scientific process. Moreover, scientists should make clear when they are addressing an issue from: a. an individual standpoint
as a scientist, b. an individual standpoint as a civilian, and c. when they are representing a scientific community (Post and
Bienzeisler, 2024; Steneck, 2011).





Based on literature, the following advocacy guidelines are included as elements in the converted advocacy text (N.B. not all alterations are included in this list, but one or two examples per guideline):

1. Addressing the issue from the individual scientist perspective: "*As a scientist, I believe that …*"
2. Stating what should be done: "*So my message to everyone is: get rid of those garden tiles!*"
3. Uncertainty in science: "*Calculating the precise impact of climate policy is difficult. But all in all …*"
4. Addressing the issue from scientific community perspective: "*While other research shows that …*"
5. Addressing the issue from individual civilian perspective (personalization): "*I recently removed all the tiles from my garden*" and "*That is why I also built a wadi myself*"

These five types of additions result in the Dutch Issue Advocate text being 425 words; see Appendix A for a translated version of the text. The title of the Issue Advocate text is the same as that of the Science Arbiter text.

### 3.3 Visual element conversion

To assess the effect of the visual on the credibility of the text and the author (as hypothesised by Leerink et al., 2024), we add two types of visual element: an Excel bar chart providing data on the temperature effect when everyone in a list of five Dutch
cities would make their garden green; and a stock photo of a wadi (taken from the website of a garden designer), intentionally without any people in the photo to make it as general as possible. The bar chart is a slight adaptation of one of the visual elements in the original KlimaatHelpdesk article and is intended as a 'scientific' visual that we (following Hypothesis 2) expect would increase the credibility of the article and thereby the author.

### 3.4 Respondents and survey design

Differences in effects between the condition texts and visual elements are measured with a survey. Questions to assess whether respondents pick up on advocacy in the text, as well as the credibility measure, are adapted from Kotcher et al. (2017) and Leerink et al. (2024). Data has been collected through Ipsos I&O, one of the largest public survey panels in the Netherlands, from 7 January 2025 to 21 January 2025. In total, 1,002 respondents filled out the survey. The respondents were recruited from the national Ipsos I&O panel, and were representative for the wider Dutch adult population.

At the start of the survey, respondents read and agree to a consent statement, after which they answer questions to assess their Science Capital & Trust and their level of Climate Change Concern. The respondents are then randomly assigned to either the Science Arbiter or the Issue Advocate text condition (Appendix A) and the Bar chart or Photo condition (Appendix B), which determine the corresponding combination of text and visual element they read. All subsequent questions are the same, irrespective of condition.

After reading the text-with-visual, respondents are asked to reflect in one or two sentences of open text on their reading or opt not to provide feedback. They cannot navigate back to the text, so must answer all questions from their memory. Six respondents report they don't know what the text is about because they hadn't read it, so these respondents are removed from the analysis. This means that the analysis is done with 996 respondents: 251 in the condition "Science Arbiter with Bar chart",



254 in the condition "Science Arbiter with Photo", 242 in the condition "Issue Advocate with Bar chart" and 249 in the condition "Issue Advocate with Photo".

## 3.5 Measures

### 3.5.1 Science Capital & Trust

The five questions on Science Capital & Trust are derived from IMPACTLAB (Peeters et al., 2022; Leerink et al., 2024) and assessed using a seven-point Likert scale. They are "I am generally aware of new scientific discoveries and developments", "I generally find scientists to be trustworthy", "I am interested in the scientific process and the results it yields", "I regularly talk about science with other people in my study, work or free time", and "I think it's important that scientists communicate about their research". The three questions on level of climate concern are "I think human influence on climate change is an important issue", "I want to know more about climate change", and "The influence of humans on climate change is exaggerated".

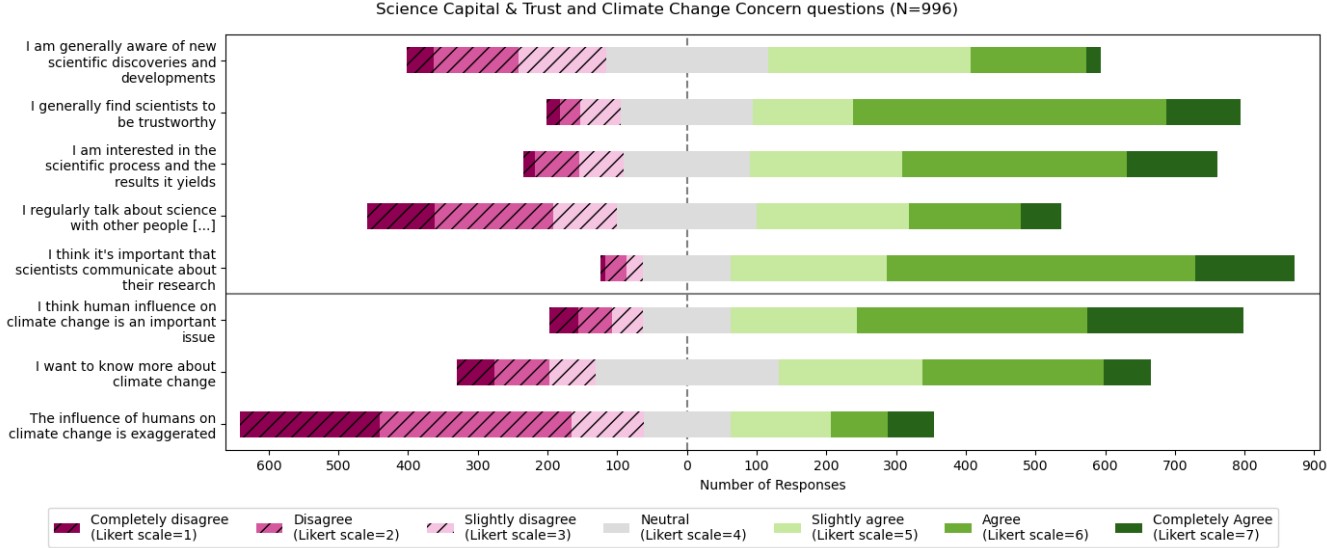

**Figure 1: The responses to statements on the Science Capital & Trust (top five questions) and Climate Change Concern (bottom three questions) of the respondents, assessed using a seven-point Likert scale. The responses are highly skewed towards high Science Capital & Trust and high Climate Change Concern.**

The average Science Capital & Trust of the respondents is relatively high (Figure 1; M=4.8; SD=1.1), with an acceptable internal consistency (Heo et al., 2015) between the five questions (Cronbach's $\alpha = 0.80$). Digging a bit deeper into the demographic background of the respondents (Appendix Figure C1), there is a strong difference ($p \ll 0.001$) in the mean score for Science Capital & Trust between education level (with respondents with a high education level scoring much higher), gender (with men scoring much higher), age group (with 25-39 scoring highest) and the political party the respondents voted on at the last election (with trust highest for left- and liberal-leaning parties).





### 3.5.2 Climate Change Concern

The average Climate Change Concern score is also high (Figure 1; M=4.9; SD=1.4), again with an acceptable internal consistency (Cronbach's $\alpha$ = 0.82). Note that, because the framing of the third question (on the exaggeration of human influence on climate) is opposite to that of the first two, the Likert score for that question is negated in the calculation of the average Climate Change Concern score. Categorising the mean Climate Change Concern score by different demographic variables (Appendix Figure C2) highlights that again there is a strong difference between education levels. The difference

between gender is not significant, and there is a relatively small (but significant) difference between age groups. As expected, the difference between political party is extremely strong, with the average Climate Change Concern score for respondents who voted for right-leaning parties much lower than that score for people who voted for left-leaning parties.

### 3.5.3 Perceived Credibility of the authoring scientist

To assess the credibility of the authoring scientist, we ask respondents to fill out nine statements about his characteristics. The

scores for these nine statements (on a seven-point Likert scale) are averaged into one construct that we call 'Perceived Credibility', following Kotcher et al. (2017) who in turn based their credibility measure on McCroskey & Teven (1999). Their original credibility measure consists of 18 statements, grouped in three factors of six statements, measuring perceived competence, integrity and goodwill of the scientist. Kotcher et al. (2017) took three statements from each factor as an adapted version of the survey: the competence of the author (whether the author is expert, competent and intelligent), the integrity of

the author (whether the author is trustworthy, honest and sincere), and the goodwill of the author (whether the author is sensitive, concerned about society and cares about society).

## 4. Results

### 4.1 Assessment of differences between conditions

To assess the degree to which our four conditions are perceived differently with respect to personalisation, the respondents are

asked – immediately after reading the text – to rate the degree to which they experienced the text as professional versus personal and formal versus informal. Figure 2 shows that the Issue Advocate text is perceived as much more personal and much more informal than the Science Arbiter text; but that the difference between the Bar chart and the Photo is much smaller. The average score was relatively neutral, between 3.75 and 4.35, in the four conditions.



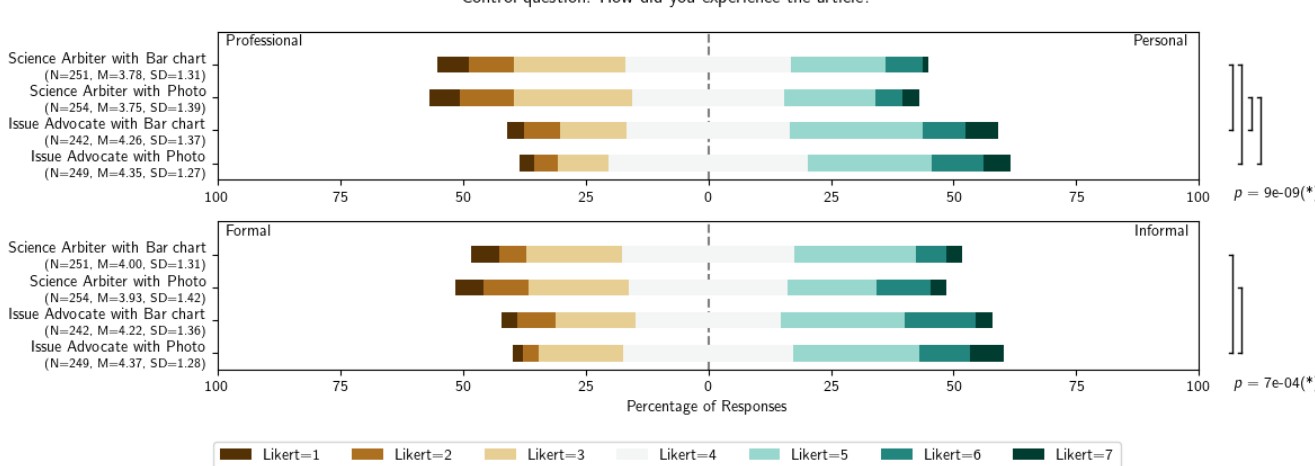

**Figure 2: The results of the control question: the degree to which the respondents evaluate the text as professional versus personal and formal versus informal; both on a seven-point Likert scale. ANOVA $p$-values are indicated on the right when $p < 0.05$, and brackets denote categories between which post hoc tests (using the Holm correction to adjust p; Holm, 1979) indicated $p < 0.05$.**

## 4.2 Perceived Credibility of the authoring scientist

For all four conditions, the average Perceived Credibility is higher than five out of seven and less than 10% of the respondents perceive the credibility as lower than three out of seven, indicating that the author is perceived as credible to very credible by a large majority of all respondents. The mean scores are between 5.1 and 5.3, with standard deviations between 0.9 and 1.0. The internal consistency of the Perceived Credibility construct in our data is high, with a Cronbach's $\alpha$ of 0.91.

While the mean scores for the two Issue Advocate conditions are slightly higher than those for the two Science Arbiter conditions and the overall ANOVA-test yields a significant difference ($p = 0.036$), the differences between the four conditions are not statistically significant after using the Holm (1979) correction to adjust $p$. We can therefore not conclude that any of the four conditions yields a higher (or lower) Perceived Credibility than the others. However, when we combine all respondents into two groups only based on the text condition (so ignoring the visual element condition), the scores for the Issue Advocate are significantly higher than scores for the Science Arbiter texts ($p = 0.005$). Combining the respondents into two groups based only on the visual element, on the other hand, does not yield statistically different results between the Bar chart and



Photo                    conditions                (                    $p = 0.521$                    ).

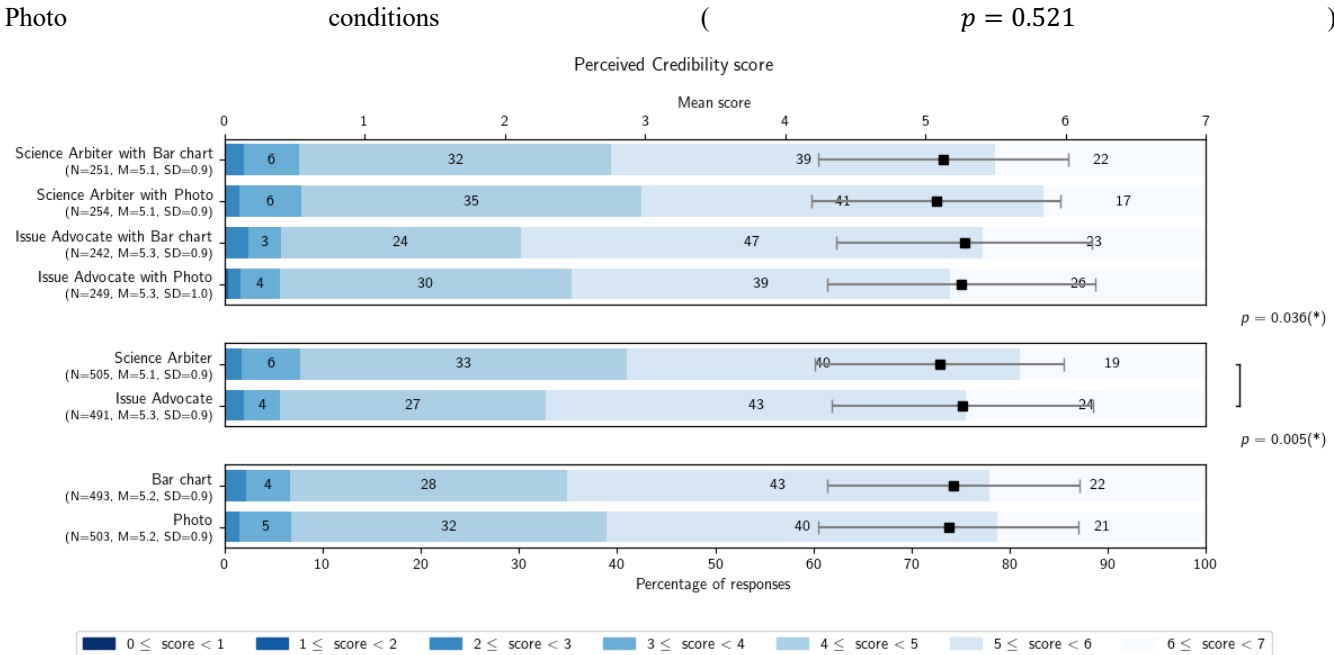

**Figure 3: The Perceived Credibility scores, as the average of nine statements about the author on a seven-point Likert score, for all four conditions (top), the two text conditions (middle) and the two visual conditions (bottom). The black squares give the means, with the whiskers indicating the standard deviations. The coloured sections in the bars indicate the fraction of respondents that gave**
**a Perceived Credibility score in each of seven ranges (note that because of the averaging, the scores are not integers anymore), with the text indicating the percentages in each range. While the ANOVA $p$-value comparing all four conditions is significant ($p < 0.05$), none of post hoc tests between conditions (using the Holm correction to adjust p; Holm, 1979) indicates $p < 0.05$. The ANOVA $p$-value comparing only the text conditions is significant ($p = 0.005$), but the ANOVA $p$-value comparing only the visual condition is not significant.**

To explore the construct of the Perceived Credibility further, Figure 4 shows the responses to the nine questions that together make up the Perceived Credibility. The respondents rate the author overwhelmingly positive on all characteristics – except perhaps for sensitivity. For five of the characteristics (sincerity, expertise, trustworthiness, competence, and care about society), there is no significant difference between the four conditions. For the other four (and especially for sensitivity) there are significant differences, and in each case the Issue Advocate texts get higher scores than the Science Arbiter texts. The

effect of the visual element is much less pronounced, with no statistically significant differences between the two visual conditions within a text condition.





**Figure 4: The results of the nine statements that combine to Perceived Credibility; all on a seven-point Likert scale. The statements are grouped by the three dimensions of credibility: Competence (blue backgrounds), integrity (orange backgrounds) and goodwill (green backgrounds). ANOVA $p$-values are indicated on the right when $p < 0.05$, and brackets denote categories between which post hoc tests (using the Holm correction to adjust p; Holm, 1979) indicates $p < 0.05$.**





**Figure 5: The results of the four goals questions; all on a seven-point Likert scale. ANOVA *p*-values are indicated on the right when $p < 0.05$, and brackets denote categories between which post hoc tests (using the Holm correction to adjust p; Holm, 1979) indicated $p < 0.05$.**

## 4.2 Goals and effect of the text

After answering the question about the credibility of the authoring scientist, respondents are asked to answer a set of four questions about the goals of the text. Figure 5 shows that most respondents find that the text provides neutral information, although the fraction of respondents that give a relatively low score for this item is significantly higher in group that has read the condition of the Issue Advocate with Photo than in the group that has read the Science Arbiter with Bar chart condition.



The second item reveals that almost all respondents feel the text is based on scientific research, and that there is no significant difference between the four conditions. There is a clear difference in the third item though, about whether the article mostly provides the author's opinion: while most respondents were neutral or did not agree with the statement, the fraction of respondent that agree with that statement is much higher in the two Issue Advocate conditions. Finally, slightly more

respondents indicate that the visual element fosters understanding the text, but this does not differ significantly between the groups that have seen the Bar chart or the Photo.

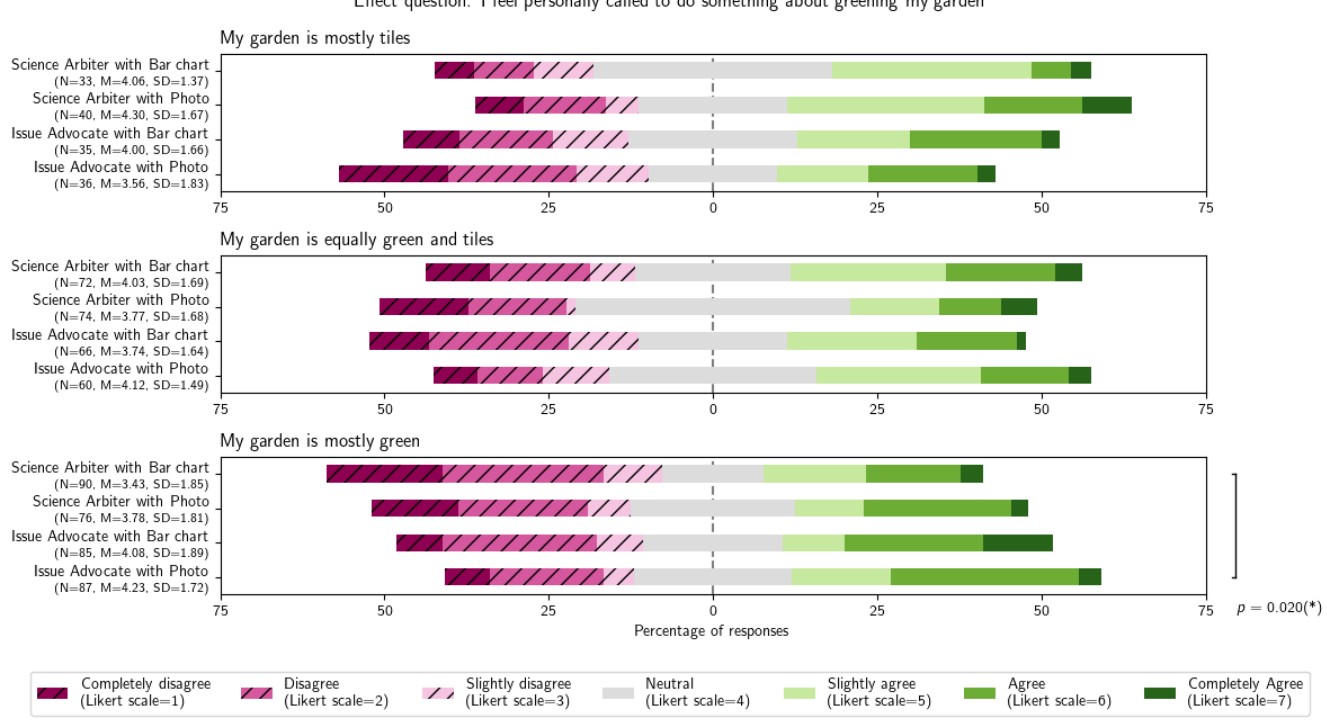

**Figure 6: The results of the action question; on a seven-point Likert scale. Only respondents that indicate they have a garden are asked to fill out this question, and only after we ask them to what extent their garden is green (the three categories of the subplots).**
**ANOVA $p$-values are indicated on the right when $p < 0.05$, and brackets denote categories between which post hoc tests (using the Holm correction to adjust p; Holm, 1979) indicate $p < 0.05$.**

Finally, we ask the respondents to what extent they now feel called to do something about greening their garden. Because we want to include only respondents for whom greening their garden is a realistic option, we first ask them whether they have a garden at all and, if so, to what extent that is green. This means that the number of respondents in each of these categories is

much lower than the approximately 250 for the other questions. The statistical power is hence also lower, and this explains why the only (small) statistically significant difference is between respondents that have seen the Issue Advocate with Photo and the Science Arbiter with Bar chart conditions, but only for those that indicate they already have a mostly-green garden. The extent to which they can therefore further green their garden is limited.





There is also some indication that respondents with a garden that is mostly tiled (upper panel in Figure 6) feel more called to
green their garden when they read the Science Arbiter text, but the difference between the conditions is not statistically
significant, even when the two types of visual elements are lumped together.

## 5. Conclusions and Discussion

The goal of this study is to evaluate whether advocacy in texts about climate change impacts and solutions influences the
credibility of the authoring scientist as perceived by the public. We evaluate possible differences between two conditions,
based on Pielke's (2007) Science Arbiter and Issue Advocate roles, using an online survey.

### 5.1 Validity of text conversion and credibility measure

Our results indicate that adding advocating elements does not undermine the (perceived) scientific underpinning of the text,
which is a core objective for KlimaatHelpdesk (Stichting KlimaatHelpdesk, 2023a). These results also show that whether a
text is perceived to be shaped by personal views of the author, and whether it is perceived to be based on scientific evidence,
are not mutually exclusive in the sense that a text can be both scientifically underpinned *and* shaped by personal views.
As an alternate view, the distribution of answers to the statement about the text being shaped by personal views (third panel of
Figure 5) is notably more widespread than the distribution of answers to the statements about the article providing neutral
information and being based on scientific research (first two panels of Figure 5). Although the difference between the
conditions is perceived as intended, less than 40% of the respondents agree that the text is shaped on personal views even in
the Issue Advocate condition. Similar results were found by Leerink et al. (2024), who included a condition with a 'highly
personalized' text, which was considered to be based on personal views by only around 50% of respondents in that condition.
This means that either the converted text could have been manipulated even more strongly to have respondents perceive it as
shaped by personal views more strongly, or that a text that is perceived as being very much based on scientific evidence is
inherently *not* seen as very much based on personal views. In that sense, maybe the two *are* mutually exclusive. Future research
may investigate this dynamic by experimenting with extremer manipulations in the levels of scientific underpinning and
shaping by personal views and testing different combinations.

### 5.2 Findings

The first research question in this project is as follows: *How does a text written in the Issue Advocate role versus the Science
Arbiter role affect the credibility of the writing scientist?* Looking at the credibility measure, Perceived Credibility of the author
increases by including a level of advocacy (Issue Advocate) as opposed to a text written in a neutral tone (Science Arbiter)
when we neglect the visual element condition. These outcomes are in line with previous research (Cologna et al., 2021; Kotcher
et al., 2017).





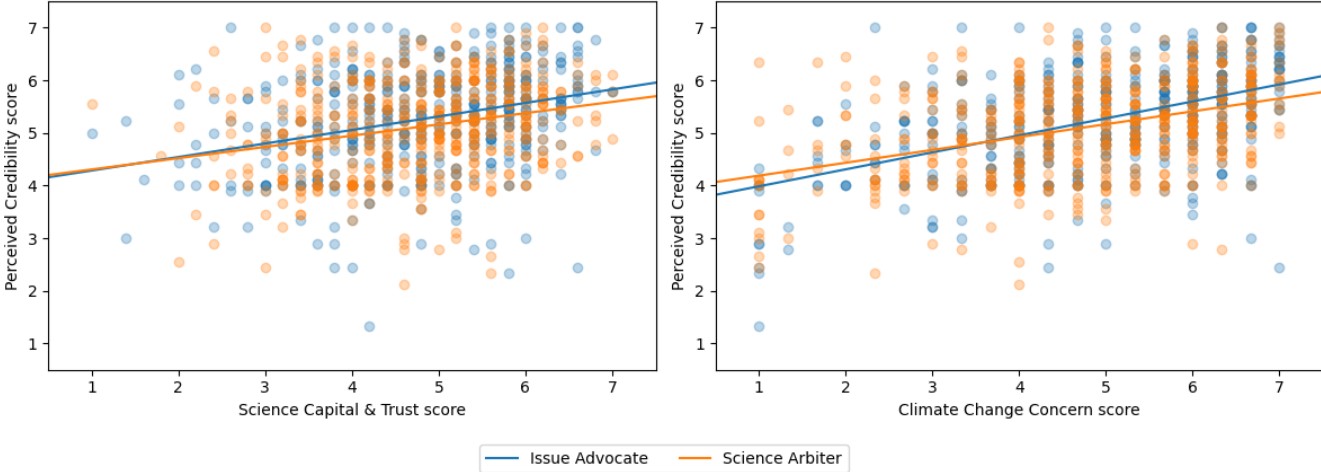

**Figure 7: Scatter plots of the relation between the Science Capital & Trust (left) and Climate Change Concern (right) scores and the
Perceived Credibility score, for each of the respondents, coloured by the text condition that they read (orange for Science Arbiter
and blue for Issue Advocate). Lines give best fits through each of the data sets for the two text conditions.**

Notably, the confidence interval on the mean credibility is quite large (SD = 0.89 for Science Arbiter and SD = 0.93 for Issue
Advocate), which might reflect polarization in the climate change debate (Donner, 2014; Post and Bienzeisler, 2024). To
confirm whether this polarization appears in our data, we combined the pretest scores of Figure 1 with the Perceived Credibility
scores of Figure 3. As shown in Figure 7, the best-line fits are steeper for the Issue advocate texts than for the Science Arbiter
texts.

Performing an Ordinary Least Squares regression fit on the data in Figure 7 reveals that the interaction term between Perceived
Credibility and Science Capital & Trust are not significant ($p = 0.401$) but that the interaction term between Perceived
Credibility and Climate Change Concern is significant ($p = 0.029$), indicating that the steepness of the two lines in the right
panel are significantly different. This suggests a (small) polarising effect of the Issue Advocate condition, where respondents
with a relatively low Climate Change Concern score perceive the credibility of the author relatively lower in the Issue Advocate
text than in the Science Arbiter text, while respondents with a relatively high Climate Change Concern score perceive the
credibility of the author relatively higher in the Issue Advocate text than in the Science Arbiter text.

In line with the results we find here, previous studies found that credibility of scientists is increased when they are perceived
to promote the well-being of others (Capstick et al., 2022) and when they are perceived to act in the interest of society (Cologna
et al., 2021). Higher scores on the goodwill domain may also indicate that the Perceived Credibility of the author can increase
when readers are convinced that the author has a personal passion for the topic and therefore acts sustainably themself. Such
results were found by Attari et al. (2016), who showed that a scientist's perceived credibility is greatly reduced when the
carbon footprint of the scientist themselves is (allegedly) high. They furthermore found that such a high footprint can strongly
affect both the readers' intentions in changing personal energy consumption (Attari et al., 2016) as well as the readers' support
for policies that the scientist advocates (Attari et al., 2019) in negative ways. Attari et al. conclude that scientists receive





support when they 'practice what they preach'. In our Issue Advocacy text, this is the acknowledgement of the authoring scientist that he had removed the tiles from his own garden. This effect of personal behaviours of the scientist on their credibility is also found to be something that some scientists themselves worry about and which keeps them from engaging in advocacy,

because they feel their own carbon footprint is too high (Dablander et al., 2024).

Furthermore, the changes in the Issue Advocate version of text have also increased the narrativity of the text, which may also have helped increase the Perceived Credibility. Yang and Hobbs (2020) showed that a text about gene editing was found more credible when it had a higher degree of narrativity. Figure 2 shows that the Issue Advocate texts are much more personal, and following Leerink et al. (2024), a text with more personalisation is more interesting to read.

The effect of visual element on the Perceived Credibility is low. The answer to our second Research Question ("*How does a visual element that shows scientific data affect the credibility of the author, compared to a more general stock photo*") is that there is no statistical difference between the two. Our second hypothesis is therefore rejected. Although past research showed potentially beneficial effects of various types of visuals in both science communication in general and climate communication specifically, we did not find big effects, not differences between the two types of visuals.

It may be that our photograph was not of sufficient quality (see Zhu et al., 2021, on the importance of high-quality photographs to affect affective and cognitive aspects of communication), or that our source attribution for the visuals was not sufficiently clear (see Li et al., 2018, on how source attribution may serve as a peripheral cue). It may also be that both types of visuals used in our experiment were missing people (see León et al., 2022, on the importance of showing "real" people or impacts or actions by people who are directly affected). Although we purposefully did not include people in our visuals to prevent

potential indirect effects due cultural, political, or stereotypical effects, this may have invertedly also limited their effectiveness. It may also be that perceived credibility did not differ across the types of visuals because both types can be congruent with the central message conveyed in the text. Recent research on text-visual congruency shows mixed results (see for example Mosallaei and Feldman (2024) and O'Neill et al. (2023) on how perceived congruency may affect information processing).

### 5.3 Limitations

The data on the Perceived Credibility measure reveals a ceiling effect, especially in the distribution plots of the individual statements (Figure 4): nearly all statements show a strong left-skewness, with a substantial number of answers in the extreme on the right (positive characteristics of the scientist) and close to none on the left (negative characteristics). The same dynamic shows in the distribution of the average perceived credibility for each respondent (Figure 3). This ceiling effect may be partly

due to the non-controversial nature of greening of gardens, or that mostly people enthusiastic about gardening accepted the invitation to the survey. As also found by Kotcher et al (2017), advocacy on more controversial topics – such as nuclear energy in their case – may decrease the perceived credibility of a scientist. It would thus be interesting to repeat our study with a text on a more controversial topic.



Furthermore, we cannot directly link the authoring scientist's perceived credibility to that of scientists in general. Even though
all texts started with a byline stating that the author of the text was "Prof. Ben van Weel (Utrecht University) – Professor of
Ecology and Climate", we did not explicitly ask whether the author was perceived as a scientist, so do not know to what extent
the respondents have appreciated that the author himself was a scientist.

Finally, the survey done here is an isolated event, performed within a panel that does regular surveys, and is in that sense a lab
setting, even though we used realistic stimulus material based on an existing website. The results in such lab settings may not
necessarily be translated to real-world situations, which means that additional field studies would be valuable (Grzyb and
Dolinski, 2021).

## 5.4 Implications & future research

The results of this study are in line with previous research (Cologna et al., 2021; Kotcher et al., 2017), indicating that scientists
can engage in advocacy without losing credibility and may even experience an increase in Perceived Credibility in the goodwill
domain. Given the large number of scientists willing to advocate, but one of the concerns being this possible loss of credibility
(Dablander et al., 2024), our results may empower willing scientists to take the next step and start speaking out on issues they
find important. Advocacy could thus provide scientists with an outlet for what they are passionate about or feel strong urgency
about. If advocacy can indeed inspire both individual behaviour change as well as influence policy making, their speaking out
may also contribute to the dealing with climate change on a grander scale.

However, whether scientists' advocacy indeed has such effects is still an understudied topic (Wijnen et al., 2024), although
some previous research can give a direction for future research. Still, these studies focused more on the effect of the (perceived)
behaviour of the author than the implications of advocacy by the author. This highlights the need of research into the effect of
advocacy on individual intended and actualized behaviour. In addition, as advocacy may also aim to influence policy-making,
it is also relevant to study to which extent scientists' advocacy can have an influence on governmental or organizational levels.

**Acknowledgements**

Funding for this project was provided through an Agnites Vrolik Award by the Utrecht University Fund. We thank Thijs
Beirnaert from De Knoest Tuinen in Tilburg for permission to use his photograph of the wadi.

**Code/Data availability**

The stacked bar graph plots were made using the plot_likert library, distributed under a BSD-3 license at
https://github.com/nmalkin/plot-likert/. In the spirit of Open Science, all data and scripts used for the manuscript are available
at https://doi.org/10.5281/zenodo.15755647.



**Author contributions**

EvS designed the survey, analyzed the data from the survey and wrote the draft of the manuscript. CW designed the first pilot of the survey and wrote a thesis version of the manuscript. All authors designed the study and edited the manuscript.

**Competing interests**

EvS is an ambassador for the KlimaatHelpdesk.

**Ethical statement**

The research proposal and protocol for this study was approved by Utrecht University's Science-Geo Ethics Review Board (ERB Review Science-24-0139) on 20 December 2024.

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



**Appendix A**

Below is a translated version of the text used in this study. Lines that were only included in the Issue Advocate version are *italic and underlined*.

**Removing all tiles from Dutch gardens is good for climate**

*By Prof. Ben van Weel (Utrecht University) – Professor of Ecology and Climate*

*I recently removed all the tiles from my garden.* The design of gardens has an effect on the climate in a city. Trees, grass and ponds are all more effective in lowering the temperature than tiles. If all gardens were completely greened, it would be about 0.5 °C cooler in large cities; *that is a lot*. In addition, green gardens reduce other climate problems such as flooding and drought. *So my message to everyone is: get rid of those garden tiles!*

Our research shows that it can be about 2 °C warmer in Dutch cities at night than in the surrounding countryside. On very hot days, it can even be 5 °C warmer. We have discovered that 10% more green surface in the city can lower the air temperature by about 0.5 °C (Figure 1). Gardens with a lot of greenery and few tiles contribute to this.

*At the moment, there is no national policy to green gardens. While* other research shows that Dutch city gardens consist of an average of 36% vegetation. But that percentage varies greatly between cities. Based on the measurements, we can estimate
that the temperature in some cities can drop by half a degree if all city gardens are greened.

*As a scientist, I believe that the government should use this knowledge to steer more towards greening gardens. I think that financial aid, for example, can persuade people who want to green their gardens but have not yet done so.*

**Effect on reducing climate problems**

A green garden can also contribute to reducing climate problems such as flooding. The soil in greener gardens can retain more rainwater than the tiles in gardens with more stones, which means that less water flows off into the sewer. Research in a district in Utrecht shows that 15% more green area in the garden can ensure 24% less water drainage into the sewer.

Green gardens also help to reduce drought. For example, a "wadi" in the garden can collect rainwater for longer, which means that more groundwater is replenished. The garden needs that groundwater during long dry periods to stay alive. *That is why I*
*also built a wadi myself.*

*Calculating the precise impact of climate policy is difficult. But* all in all, research shows that greening gardens is an easy way to reduce the effects of the climate crisis in Dutch cities.



**Appendix B**

Below are the two translated visual elements used (the bar chart and the photo), and their caption.

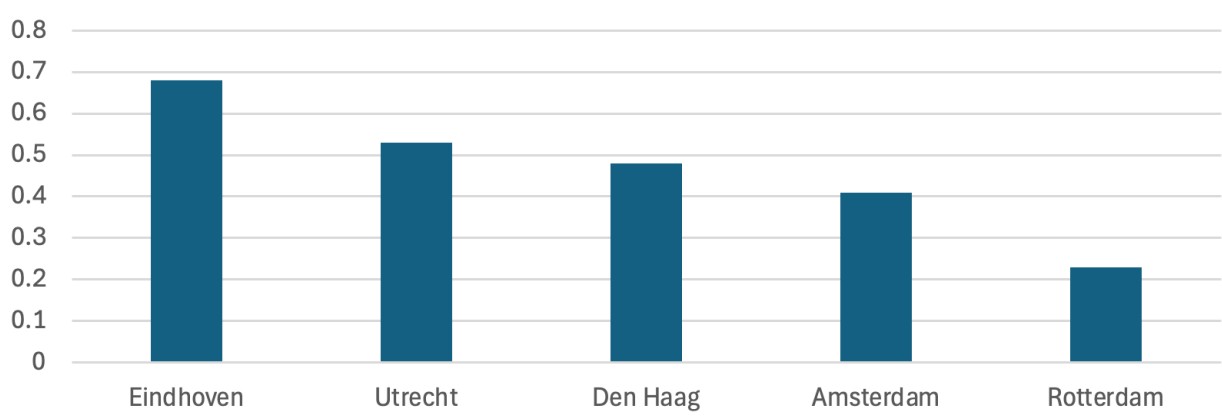

**Figure B1: The effect on air temperature if all gardens in five major cities are greened.**

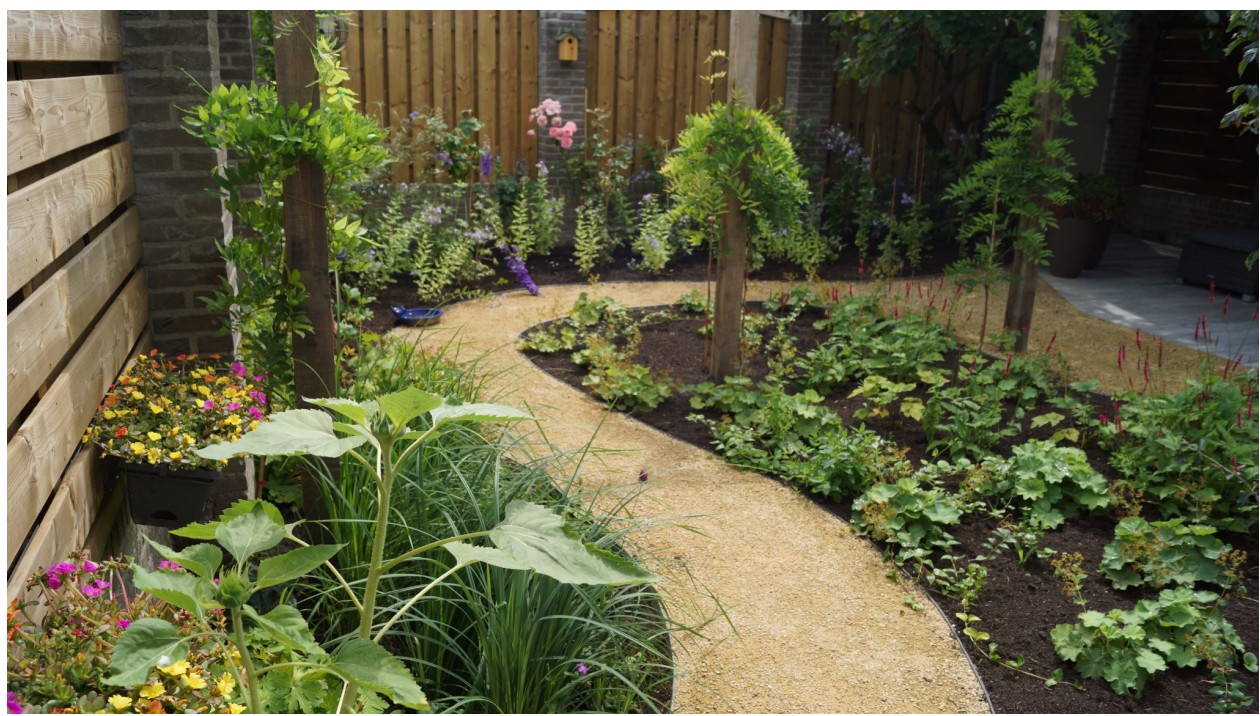


**Photo B2: A garden with a landscaped wadi**



**Appendix C**



**Figure C1: The mean Science Capital & Trust score of the respondents, separated by some of their demographic information. The political parties have roughly been ordered from right-leaning (top) to left-leaning (bottom). ANOVA $p$-values are indicated on the right when $p < 0.05$, and brackets denote categories between which post hoc tests then indicate $p < 0.05$.**



**Figure C2: The mean Climate Change Concern score of the respondents, separated by some of their demographic information. The political parties have roughly been ordered from right-leaning (top) to left-leaning (bottom). ANOVA $p$-values are indicated on the right when $p < 0.05$, and brackets denote categories between which post hoc tests (using the Holm correction to adjust $p$; Holm, 1979) then indicate $p < 0.05$.**