# Peer review of "The effect of advocacy on perceived credibility of climate scientists in a Dutch text on greening of gardens"

_EGUsphere, 2025_

## Author Comment (AC1)

**Reviewer #1**

In this study the authors have investigated how the role of the scientist (arbiter versus advocate) as well as illustration (photo versus bar chart) used influences the perception of the reader. I find this study to be interesting and well-presented.

We thank the reviewer for these very encouraging and warm comments; we appreciate their support for our study.

Apart from one potential typological error (is the sentence on lines 254-255 cut short?), I found the study to be carefully prepared and clear. However, I do have some critical comments.

Indeed, there was a formatting issue just above Figure 3, which meant the last words of that sentence ended on the top of page 10 of the original pdf. That has now been fixed.

The type of the text used in the study plays a critical role. Here, the focus is mainly on asking individuals to manage their gardens differently, hence focusing on the responsibility of the individual. Would the reception of the respondents have been different if the text would have not been mainly focused on action that takes place on their yards, but calling for e.g. parking lots to be partially greened or city planning to include more green space when building new housing? Also, the last sentence of the arbiter text can be questioned, if it was a good choice. The (imaginary) professor is offering a recommendation (although a very light one) beyond their field, which might reduce credibility. For sure this does not discredit the whole study, but I do find that the comparison would have been better without the last sentence, in order to answer the research question posed here.

The reviewer has a good point here. Unfortunately, we can't redo the entire experiment. We have now added a paragraph about the advocacy beyond the scientist's expertise in the potential limitations to the discussion section (lines 421-423 of the track-changed pdf):

"On the other hand, the authoring scientist offers a recommendation in the Issue Advocate text outside their immediate field of expertise ("I think that financial aid, for example, can persuade people who want to green their gardens but have not yet done so"), which in hindsight might reduce the credibility of the author."

Unless I misunderstand the analysis, I think the point about polarisation (line 345) requires some more nuance in the discussion. There are many reasons why people who are not concerned about climate issues might feel the need or desire to discredit a scientist who more actively promotes action. Unless I am misinterpreting, the authors suggest causality,

and I do not think this is necessarily the case. Rather, the response is or at least can be fully reactionary to the theme.

This is a good point by the reviewer. We have now added that note to the revised manuscript (lines 378-380 of the track-changed pdf):

"Note that this small effect does not necessarily imply a causal relation between level of activism and polarisation; it could for example also be that the response is reactionary to the theme."

Furthermore, and this applies to all of the results, I would recommend also some discussions on the longer-term effects. After the immediate — potentially emotional — reaction to the different texts, their impacts may be different long-term, through e.g. sensitisation to similar points made in later-encountered texts.

The reviewer is right that we could have more extensively discussed that we don't have information about the longer-term effects of our experiment. We now added a paragraph on this limitation to the revised manuscript (lines 427-434 of the track-changed text):

"Since we didn't track the respondents over longer times, we do not know what the longer-term effects of our text has been. Given that the immediate effect of our intervention was already relatively small, we can imagine that the long-term effects (weeks to years) of one such intervention would be very small. On the other hand, many in the public are repeatedly exposed to climate scientist's advice or insight on society, so it could be that a continuously increased level of activism by all climate scientists would increase the perceived credibility of climate change on the long term; something to be further explored in follow-up studies."

Related to the above point about long-term effects, I would also welcome some discussion on complementarity of the different approaches to science communication. For instance, what would happen if a very profound and (seemingly) neutral text, with some infographics on facts (arbiter) would be provided first, followed by the more personal call to action? How would this influence the perception of the public? The value of different approaches is also recognised in Pielke's book on the honest broker (p. 7). I think this point of view would enrich the discussion.

This is a very good suggestion by the reviewer. We added a section in the future research section to highlight this idea (lines 449-451 of the track-changed pdf):

"In particular, one could extend our experiment by exposing participants first to a Science Arbiter-style text, followed by an Issue Advocate-style text, to explore the complementarity of the different approaches to science communication (Pielke Jr, 2007, p7)."

---

## Author Comment (AC2)

**Reviewer #2**

The study examined whether advocacy in climate change texts affects how the public perceives the credibility of the scientist authoring them, comparing two roles: Science Arbiter (neutral) and Issue Advocate (advocating). It presents some interesting findings - with the key finding being that advocacy doesn't undermine the scientific credibility of the text and scientist.

We thank the reviewer for their supportive comments and suggestions, which we have incorporated into the revised version of our manuscript.

There are some areas in which this paper can be improved, my main concern is method section. All results presented in the section should be moved to the results sections. As it stand the data analysis section is not well structure. Removing results from this section would give room to cleary explain how the Measures in 3.5 where adapted to suite this study and answer the hypothesis of the study.

We agree with the reviewer that our discussion of the measures could have been clearer. In the revised manuscript, we have moved the results from the old section 3.5 (Measures) to the Results section and renamed it Section 4.1. We have further expanded section 3.5, and moved the Perceived Credibility description to a new section 3.6 (since it is not technically a measure).

**My specific points are as follows**

Abstract L10- The opening statement would benefit with an expansion of why climate scientists rephrain from advocacy

We feel it is already clear in the first sentence that climate scientists refrain from advocacy "because they worry it decreases their credibility". We don't see how we can further clarify this, while keeping the abstract concise and to-the-point.

Introduction L26- examples of consequences would give a clearer picture to the reader.

In the revised manuscript, we have now added the "fear of repercussions or reprimands from their institutes or peers" as examples of negative consequences (lines 21-22 of the track-changed pdf).

There are a lot typos than need addressing

We have done a careful rereading of the manuscript and removed all the typos we found. If any typos remain, we are confident that the GC copy-editors will fix them during typesetting.

Section 2, L100 - It will important to reference other Geoscience studies which critically looked at the effects of visuals and effects of colours when lobying (e.g. Williams et al 2023, <a href="https://doi.org/10.5194/gc-6-111-2023">https://doi.org/10.5194/gc-6-111-2023</a>). Also prior experience also influences how the individual undertsands the visual. Which is in line to my other comments on the composition of the participants (a table or figure) is required to show education levels of the participants, and this need to clear how this was handled in the data analysis (Sect 3.4)

We have now added the Williams et al (2023) reference to the theory on credibility in visual elements (lines 102-103 of the track-changed manuscript).

As for the composition of the participants, we now expand section 3.4 to highlight that the participants "had a similar distribution of education levels as the Dutch public as a whole" (lines 188-189 of the track-changed pdf). Since all our data is also publicly available, we do not feel that an extra table is required.

**Theory section should be condensed a bit more, there has been a lot of repetition.**

In the revised manuscript, we have now somewhat shortened the theory section, especially the first few paragraphs of section 2.1 (lines 58-78 of the track-changed pdf). However, we feel that the rest of the theory is important and necessary to introduce our research questions and hypotheses.

I would like to see more detailed explanation in section 3.2, with reference to the key figures in the Appendix. This link has been missing throughout the Method section. For example in section 3.3 the authors mention about the excel bar chart I looked for it but couldn't, then i saw it in the appendix. Please make reference where appropriate.

The reviewer has a very good point here; in the original manuscript we omitted explicit references to the Appendices in sections 3.2 and 3.3. We have now added these in the revised version of the manuscript (lines 157, 165, 179 and 180 of the track-changed pdf).

---

## Author Response (AR1)

**Editor comments following peer-review**

1. Authors response to Reviewer #1.

I thank the authors for their clear response to the comments by Reviewer 1 and for thoughtfully incorporating their recommendations to improve the manuscript.

2. Authors response to Reviewer #2.

I thank the authors for their clear response to the comments by Reviewer 1 and for accepting to incorporate the structural changes to the Methods (3.5) and Results sections to improve the readability of the manuscript.

3. Editor's comments

The topic of the manuscript is timely and highly relevant not only to the climate science community, but to the broader discussion on public perception of (advocating) scientists.

The manuscript is very well-written and easy to follow. The research questions and hypothesis are clearly stated from the beginning and addressed in the discussion. The structure is well-developed with sections for introduction, methods, results, limitations and future work. The figures are clear and include the relevant statistical data.

In addition to the comments by Reviewers 1 and 2, I would like to contribute with minor suggestions for your consideration, as well as technical edits (mostly typos), to further strengthen the paper (see below).

We thank the editor for these very encouraging and warm comments; we appreciate their support for our study.

Discussion suggestions

To continue the discussion on Perceived Credibility and Climate Change Concern from Reviewer 1 on (lines 345-348), if I may add another point to this line, line 387-388, and Figure 7: Could it also be a case of confirmation bias? (ie. Those with high Climate change concern would rate the advocating scientist as more credible, and vice versa).

This is a very good point. We have now added another sentence to the revision (lines 383-384 of the track-changed pdf): "*Furthermore, there could also be some confirmation bias where respondents with high Climate Change Concern would rate the advocating author as more credible, and vice versa.*"

Section 3.4 on Respondents and survey design. Related to Reviewer 2 comment on the "composition of participants", I would kindly ask the authors to include a description of the demographic information asked in the survey (perhaps between lines 180-184), as this is not

explicitly mentioned until the analysis of Figure 1 (currently line 210, but I assume will be moved to Results in the final edited version).

We have now added a sentence at the beginning of section 3.5 (lines 205-206 of the track-changed manuscript: "*As demographic information was readily available as respondents were part of an existing Ipsos I&O panel, these were not measured (see Appendix C for an overview).*"

Line 75: "do no engage in advocacy, despite their willingness". If there are reasons/hurdles as to why they don't engage, please include some examples.

This sentence appeared in the initial version of the manuscript, and has been removed in the revised version (as asked by the reviewers). We now added "*because of intellectual (e.g., not the role of researcher or lack of knowledge) or practical (e.g., lack of time or skills) barriers*" (lines 77-78 of the track-changed pdf), but do not want to go much deeper as that would take a lot of space and readers could better read the Dablander et al article itself (Figure 3 in that paper is specifically about the barriers).

Line 136-137: It would be interested to add a line of how the platform vets experts (eg. university affiliation? Academic title?).

We have now added that the expertise is "as *identified by the topic of their PhD*" (line 141 of the track-changed pdf).

Line 224: It would be interesting to assess the impact of a Issue Advocate with a female name. There are studies showing how the general public associate "scientist" to male. Hence, I would recommend a future study that investigates perception and credibility by gender.

This is a very good suggestion. We have now added the following sentence to the Implications & future research section (lines 455-457 of the track-changed pdf): "*Furthermore, one could investigate whether an Issue Advocate text by a scientist with a feminine name would have the same response, as there are studies that show how the general public associate scientists with men (e.g., Suldovsky et al., 2019).*"

Technical edits

Line 40: typo in: "op dialogue".

We have changed to "*a dialogue*" in the revised manuscript (line 41 of the track-changed pdf).

Line 110: "visuals in a multimodal setting with texts", it may be more straightforward to remove the word "multimodal", as you are referring only to the two: text + visuals.

We have removed "*multimodal*" in the revised manuscript (line 116 of the track-changed pdf).

Line 119: typo, verb in singular: "these mostly focuses".

We have changed to "*these mostly focus*" in the revised manuscript (line 125 of the track-changed pdf).

Line 122: comma might be better than n-dash.

We have changed this in the revised manuscript (line 128 of the track-changed pdf).

Line 158-159: consider adding () to you're a, b, c list.

We have changed to a), b) and c) in the revised manuscript (lines 165-166 of the track-changed pdf).

Line 174: typo, word from singular to plural "types of visual elements".

We have changed this in the revised manuscript (line 180 of the track-changed pdf).

Line 184: typo, exchange 'for' for 'of' in "representative for the wider".

We have changed this in the revised manuscript (line 190 of the track-changed pdf).

Line 290: word missing, "fosters understanding of the text".

We have changed this in the revised manuscript (line 323 of the track-changed pdf).

Line 369: typo, change to "nor difference".

We have changed this in the revised manuscript (line 405 of the track-changed pdf).

Appendix B

Line 600: To remain consistent with in-text reference to Bar and Photo conditions, please consider modifying title to, eg.: (B1) Bar chart and (B2) Photo conditions.

This is a good suggestion. We have now changed this to *"Figure B"* (line 639 in the track-changed pdf). Because half the respondents saw Figure B1 and half saw Figure B2, we don't want to also put the number here.

---

## Editor Decision (ED1)

**Editor comments following peer-review**

1. Authors response to Reviewer #1.

I thank the authors for their clear response to the comments by Reviewer 1 and for thoughtfully incorporating their recommendations to improve the manuscript.

2. Authors response to Reviewer #2.

I thank the authors for their clear response to the comments by Reviewer 1 and for accepting to incorporate the structural changes to the Methods (3.5) and Results sections to improve the readability of the manuscript.

3. Editor's comments

The topic of the manuscript is timely and highly relevant not only to the climate science community, but to the broader discussion on public perception of (advocating) scientists.

The manuscript is very well-written and easy to follow. The research questions and hypothesis are clearly stated from the beginning and addressed in the discussion. The structure is well-developed with sections for introduction, methods, results, limitations and future work. The figures are clear and include the relevant statistical data.

In addition to the comments by Reviewers 1 and 2, I would like to contribute with minor suggestions for your consideration, as well as technical edits (mostly typos), to further strengthen the paper (see below).

**Discussion suggestions**

To continue the discussion on Perceived Credibility and Climate Change Concern from Reviewer 1 on (lines 345-348), if I may add another point to this line, line 387-388, and Figure 7: Could it also be a case of confirmation bias? (ie. Those with high Climate change concern would rate the advocating scientist as more credible, and vice versa).

Section 3.4 on Respondents and survey design. Related to Reviewer 2 comment on the "composition of participants", I would kindly ask the authors to include a description of the demographic information asked in the survey (perhaps between lines 180-184), as this is not explicitly mentioned until the analysis of Figure 1 (currently line 210, but I assume will be moved to Results in the final edited version).

Line 75: "do no engage in advocacy, despite their willingness". If there are reasons/hurdles as to why they don't engage, please include some examples.

Line 136-137: It would be interested to add a line of how the platform vets experts (eg. university affiliation? Academic title?).

Line 224: It would be interesting to assess the impact of a Issue Advocate with a female name. There are studies showing how the general public associate "scientist" to male. Hence, I would recommend a future study that investigates perception and credibility by gender.

**Technical edits**

Line 40: typo in: "op dialogue".

Line 110: "visuals in a multimodal setting with texts", it may be more straightforward to remove the word "multimodal", as you are referring only to the two: text + visuals.

Line 119: typo, verb in singular: "these mostly focuses".

Line 122: comma might be better than n-dash.

Line 158-159: consider adding () to you're a, b, c list.

Line 174: typo, word from singular to plural "types of visual elements".

Line 184: typo, exchange 'for' for 'of' in "representative for the wider".

Line 290: word missing, "fosters understanding of the text".

Line 369: typo, change to  "nor difference".

Appendix B

Line 600: To remain consistent with in-text reference to Bar and Photo conditions, please consider modifying title to, eg.: (B1) Bar chart and (B2) Photo conditions.